# Isobavachalcone as an Active Membrane Perturbing Agent and Inhibitor of ABCB1 Multidrug Transporter

**DOI:** 10.3390/molecules26154637

**Published:** 2021-07-30

**Authors:** Anna Palko-Łabuz, Maria Błaszczyk, Kamila Środa-Pomianek, Olga Wesołowska

**Affiliations:** Department of Biophysics and Neuroscience, Wroclaw Medical University, ul. Chalubinskiego 3a, 50-368 Wroclaw, Poland; anna.palko-labuz@umed.wroc.pl (A.P.-Ł.); maria.blaszczyk@student.umed.wroc.pl (M.B.); olga.wesolowska@umed.wroc.pl (O.W.)

**Keywords:** isobavachalcone, prenylated chalcone, multidrug resistance (MDR), P-glycoprotein (ABCB1; MDR1), model membrane, phospholipid bilayer, differential scanning calorimetry (DSC), molecular modeling

## Abstract

Isobavachalcone (IBC) is an active substance from the medicinal plant *Psoralea corylifolia*. This prenylated chalcone was reported to possess antioxidative, anti-inflammatory, antibacterial, and anticancer activities. Multidrug resistance (MDR) associated with the over-expression of the transporters of vast substrate specificity such as ABCB1 (P-glycoprotein) belongs to the main causes of cancer chemotherapy failure. The cytotoxic, MDR reversing, and ABCB1-inhibiting potency of isobavachalcone was studied in two cellular models: human colorectal adenocarcinoma HT29 cell line and its resistant counterpart HT29/Dx in which doxorubicin resistance was induced by prolonged drug treatment, and the variant of MDCK cells transfected with the human gene encoding ABCB1. Because MDR modulators are frequently membrane-active substances, the interaction of isobavachalcone with model phosphatidylcholine bilayers was studied by means of differential scanning calorimetry. Molecular modeling was employed to characterize the process of membrane permeation by isobavachalcone. IBC interacted with ABCB1 transporter, being a substrate and/or competitive inhibitor of ABCB1. Moreover, IBC intercalated into model membranes, significantly affecting the parameters of their main phospholipid phase transition. It was concluded that isobavachalcone interfered both with the lipid phase of cellular membrane and with ABCB1 transporter, and for this reason, its activity in MDR cancer cells was presumptively beneficial.

## 1. Introduction

Isobavachalcone (IBC), together with several other prenylated flavonoids, was identified among the active substances isolated from the medicinal plant *Psoralea corylifolia* Linn. [1]. The plant, known under the names Bakuchi, Babchi, or Bu Gu Zhi, is commonly used in Ayurvedic and traditional Chinese medicine to treat various skin problems, such as leukoderma, alopecia areata, psoriasis, and others [2]. IBC is a chalcone, prenylated at position 6 of ring A. IBC has been previously reported to possess antiviral [3,4], antibacterial [5,6], and antifungal properties [7]. It is also an active anticancer agent [8,9], inducing apoptosis and inhibiting the Akt and Erk signaling pathways [8,10]. Significant antioxidative properties of IBC [11,12] are likely to be responsible for its anti-inflammatory [13,14] and neuroprotective activities [15,16].

Multidrug resistance (MDR) is a complex network of cellular adaptations that lead to the ability of cancer cells to evade chemotherapy. MDR may be intrinsic, but most frequently it appears as a result of chemotherapy, leading to the ineffectiveness of the treatment. One of the most frequent mechanisms of MDR is an over-expression of the transporters of wide substrate specificity by resistant cancer cells that leads to the decrease of the drugs’ intracellular concentration below the killing threshold [17]. ATP-binding cassette (ABC) transporters such as ABCB1 (MDR1, P-glycoprotein), ABCC2 (MRP1), and ABCG2 (BCRP) are most frequently engaged. These are membrane-embedded proteins that gain energy from ATP hydrolysis to effectively pump xenobiotics (including drugs) out of the cell [18]. The use of inhibitors of ABC transporters (MDR modulators) has been proposed as a strategy, giving hope to improve the outcome of chemotherapy [19,20]. In spite of the lack of clinically significant outcomes up to now, new and effective modulators of ABC transporter activity are still being explored [21]. Prenylated flavonoids have been already reported to be effective ABCB1 inhibitors [22,23].

Because the binding pocket of MDR-associated transporters from the ABC subfamily is formed by amino acids from several α-helical fragments spanning cellular membrane [24,25], the interaction with the substrates occurs within the lipid milieu [26]. Additionally, the functioning of ABC transporters can be affected by membrane features, such as lipid composition and fluidity [27,28]. ABCB1 substrates and inhibitors are typically amphiphilic compounds, and as such, they possess the ability to interact with lipid bilayers. It has been proposed that anti-MDR activity and the extent of perturbation of the lipid phase of the cellular membrane by a chemical may be correlated [29]. The antibacterial activity of IBC against methicillin-resistant *S. aureus* (MRSA) strains was demonstrated to be due to chalcone-induced membrane damage [5]. In studies on MDR variants of *S. aureus*, Song et al. proposed an association between the perturbation of the bacterial inner membrane by IBC and its bactericidal activity [30]. The interaction of IBC with model lipid membranes has not been studied so far. However, another prenylated chalcone, xanthohumol [31,32], as well as some prenylated flavonoids, have been demonstrated to affect the physicochemical properties of model phospholipid bilayers to a great extent [32,33].

In the present work, the activity of IBC in two cancer cell models was investigated. Its cytotoxicity, ability to change doxorubicin (Dox) accumulation and resistance to this drug, as well as ABCB1 inhibition potency were studied in drug-sensitive and resistant human colorectal adenocarcinoma cells (HT29), and in the variant of MDCK cells expressing human ABCB1 protein. Additionally, the impact of IBC on thermotropic properties of model phospholipid membranes was analyzed for the first time. Molecular modeling was employed for further characterization of IBC interaction with membranes, especially its ability to permeate the bilayer. Briefly, the chalcone was shown to interact with ABCB1 transporter, interfering with its function. It was likely to be a competitive inhibitor of the transporter. Moreover, IBC intercalated into model membranes composed of phosphatidylcholine, significantly affecting the parameters of their main phospholipid phase transition. It was concluded that IBC interfered both with lipid phase of cellular membrane and with MDR-associated transporter.

## 2. Results

### 2.1. Cytotoxicity of Isobavachalcone to Cancer Cells

The first stage of the study was the investigation of IBC cytotoxic potential in cancer cells. As a model, two pairs of cancer cells were used. Human colorectal adenocarcinoma HT29 cell line and its Dox-resistant counterpart, HT29/Dx, were chosen as an example of drug resistance naturally induced by prolonged drug treatment. MDCK cells and the variant expressing human ABCB1 protein (MDCK-MDR1) were used as a model in which the resistance was obtained artificially by the introduction of the gene of MDR-associated transporter. IBC in a concentration up to 40 μM was found to be non-toxic to both HT29 and HT29/DX cells (Figure 1A). HT29 cells were slightly more vulnerable to the tested compound than HT29/Dx cells, but the difference was not statistically significant. In the case of MDCK cells, it was observed that IBC reduced cell growth when applied in concentrations of 20 μM or higher (Figure 1B). The IC50 value for MDCK cells was 26.6 ± 3.4 μM. MDCK-MDR1 cells were less vulnerable to IBC than MDCK cells. Additionally, an interesting observation was made in MDCK-MDR1 cells. Namely, in concentrations between 10 and 20 μM, IBC significantly stimulated the growth of cancer cells, and this effect faded at higher concentrations.

This observation prompted us to set the hypothesis that the overexpression of ABCB1 transporter in MDCK-MDR1 cells was the factor responsible for the effect. Therefore, in the next series of experiments, a known inhibitor of ABCB1 transporter, verapamil (Ver) [34], was employed. When Ver (at 75 μM) was added to MDCK-MDR1 cells together with IBC, the stimulation of cell growth induced by IBC was abolished (Figure 1C). This might suggest that IBC was a substrate of ABCB1 protein and that inhibition of the function of the transporter caused more IBC to stay inside cancer cells, thus reducing the survival rate of MDCK-MDR1 cells.

### 2.2. Intracellular Accumulation of Doxorubicin

HT29/Dx cells are partially resistant to Dox [35]. This anticancer drug is characterized by high intrinsic fluorescence. Taking advantage of this fact, microscopic observations of human adenocarcinoma cells treated with IBC were performed (Figure 2). Less Dox accumulated within resistant cells compared to sensitive ones (Figure 2A,B, respectively). Additionally, in HT29 cells, Dox fluorescence was concentrated in cellular nuclei, whereas these organelles were labeled with Dox to a much lesser extent in HT29/Dx cells. The treatment of adenocarcinoma cells with a non-toxic concentration of IBC (15 μM) affected intracellular accumulation of Dox (Figure 2C,D). Intracellular fluorescence intensity was slightly reduced in HT29 cells and significantly increased in HT29/Dx cells. Moreover, Dox seemed to concentrate more in the cellular nuclei of IBC-treated resistant cells.

### 2.3. Doxorubicin Resistance Reversal

Encouraged by the results presenting the ability of IBC to increase Dox accumulation in resistant cancer cells, we checked whether this compound would be able to reduce the resistance of HT29/Dx cells to Dox. The results of the experiments on Dox cytotoxicity showed that, indeed, HT29 cells were more sensitive to the drug (Figure 3A) than HT29/Dx cells (Figure 3B). However, the treatment of resistant cells with IBC at 15 μM concentration did not change Dox cytotoxicity toward these cells, i.e., IBC did not reverse drug resistance.

### 2.4. Accumulation of Rhodamine 123

The ABCB1 protein transports a vast group of substrates, among them many fluorescent dyes. The assay based on the measurement of intracellular accumulation of one of the typical ABCB1 substrates, rhodamine 123 (Rho123), provides a convenient functional test to assess the function of ABCB1 transporter [36]. The results of the test were presented by means of the fluorescence intensity ratio (FIR) parameter which informs about the effect of the studied compound on the ratio of intracellular fluorescence of R123 accumulated in sensitive and resistant cells. Surprisingly, when the test was performed in HT29 and HT29/Dx cells, it was observed that the amount of R123 accumulated in the sensitive cells was only c.a. 5% higher than in the resistant ones. This suggested that the transport activity of ABCB1 was not likely to be responsible for the observed Dox resistance of human adenocarcinoma cells.

When the test was performed in MDCK and MDCK-MDR1 cells, it was observed that the amount of R123 accumulated by the latter cells was c.a. two times lower than the amount recorded in MDCK cells. Figure 4 presents FIR parameters recorded for Ver and IBC. Both compounds increased FIR values in a concentration-dependent manner. FIR values obtained for IBC were, however, lower than the ones obtained for Ver. It suggested that both compounds interacted with ABCB1, interfering with its transport function.

### 2.5. Interaction with Model Membrane

ABCB1 is a membrane-embedded protein, and modification of the properties of lipid bilayer is likely to affect its functioning. For this reason, the impact of IBC on thermotropic properties of model membrane composed of phosphatidylcholine was studied by means of differential scanning calorimetry (DSC). Two species of phosophatidylcholine were used to form model membranes: DMPC possessing hydrocarbon chains of 14 C atoms, and DPPC with 16C-long chains. Phosphatidylcholine is a zwitterionic, cylindrically shaped phospholipid easily forming bilayers. It is the most abundant phospholipid in the outer leaflet of mammalian cellular membrane [37], so phosphatidylcholine bilayers are widely used as simple membrane models. The parameters describing the main phospholipid phase transition of both lipid species doped with different amounts of IBC are presented in Table 1, and exemplary thermograms can be seen in Figure 5. The addition of IBC resulted in a vanishing of pretransition in both lipids. Increasing the IBC:lipid ratio caused a decrease in main phase transition temperature together with significant broadening of the peaks mirroring transition, as judged by the increase in their half-height values. Moreover, the presence on IBC in the phosphatidylcholine bilayer reduced transition enthalpy in a concentration-dependent manner. Therefore, the results described above clearly pointed to interaction of IBC with model membrane formed from zwitterionic phospholipid.

### 2.6. Molecular Modeling

For better understanding of the interaction of IBC with the MDR transporter and cellular membrane, several physical parameters characterizing the structure and reactivity of chemical compounds were computed (Table 2). IBC possessed a significant energy gap between the highest (HOMO) and lowest (LUMO) occupied molecular orbital, which pointed to its high reactivity. Dipole moment was aligned along the long axis of the IBC molecule and directed toward the prenyl group. IBC was also characterized by relatively high lipohilicity as judged by an octanol:water partition coefficient above 4. Figure 6. is a graphical presentation of molecular modeling results. When the electrostatic potential map was superposed on the optimized structure of IBC, the spatial distribution of positive and negative regions of the molecule could be observed (Figure 6B). Positive regions were centered near the H atoms of hydroxyl groups at positions 4′ of ring B and 7 of ring A. Negative regions can be seen near the O atom of the ketone group and at O atoms from hydroxyl groups at positions 4′ of ring B and 7 of ring A. The energy levels of the HOMO and LUMO define how the molecule shares its valence electrons and donates (or accepts) electrons to (from) a ligand. HOMO orbitals of IBC were localized mainly at aromatic rings and at the double bond in the bridge between the rings (Figure 6C), whereas LUMO orbitals were localized near the single bond next to the ring B, as well as near the C atom of the ketone group (Figure 6D).

The next step was to model the energy of interaction of IBC with a membrane and its behavior within a bilayer. To accomplish this task, the web tool PerMM was employed [38]. PerMM allows for the calculation of permeability coefficients (logP) and membrane binding energies for various artificial and natural membrane models. The results, and the translocation pathway of IBC through lipid bilayer, are visualized in Figure 7. It was concluded that IBC partitioned into the lipid bilayer, with an overall favorable binding energy of −4.81 kcal/mol for DOPC membrane. Tracking of the IBC transfer energy profile revealed two deep minima at the water−lipid interfaces and a maximum in the membrane center. Membrane permeability coefficients of IBC for simple artificial membrane models were higher than for more complex natural membranes. The results of the PerMM computation clearly pointed to the amphiphilic character of IBC and its significant membrane penetrating properties.

## 3. Discussion

The analysis of the cytotoxic potential of IBC in human adenocarcinoma cell lines showed that the chalcone was virtually non-toxic to cancer cells in concentrations below 40 μM. Additionally, the difference in IBC toxicity to HT29 cells and partially Dox-resistant HT29/Dx cells was negligible. Higher cytotoxicity was observed in MDKC cells, whereas MDCK-MDR1 cells were not vulnerable to IBC in the studied concentration range. Moreover, in low concentrations (below 20 μM), IBC significantly stimulated the growth of cancer cells transfected with human *MDR1* gene. It therefore seemed that the presence of over-expressed ABCB1 transporter was responsible for the observed difference in cell growth in the presence of IBC between MDCK and MDCK-MDR1 cells. Previous research indicated that substrates of ABCB1 transporter might induce expression of the *MDR1* gene encoding this pump [39]. In our former studies on simvastatin, a competitive inhibitor of ABCB1, a drug-induced increase in *MDR1* expression was indeed observed [40]. Therefore, we speculated that the expression of *MDR1* in MDCK-MDR1 cells, additionally enhanced by the presence of the potential substrate (IBC), might be responsible for the extensive efflux of the compound, leading to the stimulation of cancer cell growth. Such a stimulation was also observed in ABCB1-overexpressing human adenocarcinoma cells (LoVo/Dx) incubated in the presence of other flavonoid compounds—baicalein and luteolin [41]. Similarly to IBC, these compounds were also recognized to be ABCB1 pump substrates. In HT29/Dx, the resistance to Dox was induced by prolonged exposure to this drug [35]. Overexpression of the following transporters ABCG2 (BCRP) [42], ABCC3 (MRP3) [35], and, at the lower level, of ABCB1 (P-glycoprotein) and ABCC2 (MRP2) [35,42], was observed in HT29/Dx cells in comparison to HT29 cells. The investigation of ABCC1 (MRP1) expression in Dox-resistant cells brought contradictory results [35,42]. Hence, it is probable that the Dox resistance in the HT29/Dx cell line was multifactorial rather than solely dependent on ABCB1 transporter activity.

IBC has been previously demonstrated to exert anti-proliferative activity in many cancer cell lines. It was shown to induce cell death via an apoptotic pathway in ovarian, lung, breast [8], gastric [10], prostate [8,43], liver [44], and colorectal cancer [45] as well as in neuroblastoma [46] and leukemia [47]. IC_50_ values recorded for the chalcone strongly depended on the cell line used and ranged between 5 and 75 μM. Previous studies on the cytotoxic potential of IBC in colorectal cancer cells yielded IC_50_ values for IBC of c.a. 50 μM in SW480 [45] and above 75 μM in HCT116 cells [45]. The IC_50_ value for IBC in HT29 cells has not been previously reported; however, the present results demonstrating IBC to be non-toxic to these cells up to 40 μM seem to be in agreement with previous reports on IBC cytotoxic potential in colon cancer cells. In some studies, IBC also demonstrated some degree of selectivity between cancer and normal cells [46,47], but in other studies similar cytotoxicity of the chalcone toward cancer and non-cancer cells was observed [8]. When Kuete et al. [9] studied the cytotoxicity of IBC in various MDR cancer cell lines, they observed lower sensitivity of ABCB1-overexpressing leukemia cells to IBC than of Dox-vulnerable cells. On the other hand, ABCG2 over-expressing breast adenocarcinoma cells were hypersensitive to the chalcone as compared to their parental cells.

Observation of differential cytotoxic activity of IBC in MDCK and MDCK-MDR1 cells prompted us to perform experiments in which transport activity of ABCB1 was inhibited by Ver. Because the presence of Ver abolished IBC-induced stimulation of cell growth, it was concluded that IBC was likely to be transported by ABCB1. The blockade of the proper function of the transporter might increase the intracellular concentration of IBC and thus reduce the viability of MDCK-MDR1 cells.

Further experiments brought clear evidence that IBC was able to increase intracellular accumulation of Dox within resistant HT29/Dx cells. It suggested that the studied compound blocked the mechanism responsible for moving the anticancer drug out of resistant cancer cells. On the other hand, IBC was demonstrated not to be able to reverse Dox resistance in HT29/Dx cells. That might have been due to the low concentration of IBC chosen for this experiment (15 μM). Another reason for IBC not being an MDR modulator in HT29/Dx cells might be the fact that the presence of functional ABCB1 protein was not the main factor deciding their survival in the presence of the anticancer drug.

This supposition was additionally supported by the results of a flow cytometric test applied to monitor the transport function of ABCB1 protein. HT29/Dx seemed not to possess measurable activity of ABCB1 transporter, in contrast to MDCK-MDR1 cells. In the latter experimental setting, IBC was demonstrated to interfere with the activity of the transporter; however, its inhibitory potency was lower than that of Ver. A similar observation was previously made for the prenylated flavonoid 8-prenylnaringenin, which was able to reduce Dox accumulation and inhibited ABCB1 activity in Dox-resistant colorectal adenocarcinoma cells (LoVo/Dx), but was not able to affect Dox cytotoxicity [23]. Taking these results together, IBC was expected to interact with ABCB1 transporter, most probably being its substrate and able to act as a competitive inhibitor in relation to other substrates of lower affinity to the transporter. Similarly, the use of the potent ABCB1 inhibitor, Ver, suppressed the activity of IBC. Additionally, as judged by the increase of Dox accumulation within HT29/Dx cells in the presence of the chalcone, IBC was also likely to interfere with some other (non-ABCB1) Dox transporting activity in these cells.

ABCB1 is an integral protein spanning the membrane. It has been shown that the composition and physical state of the membrane affects ABCB1 function [28]. It is also commonly accepted that recognition and binding of its substrates occurs within the phospholipid bilayer [26]. Many MDR modulators are membrane-active compounds, and it has been suggested that their ability to perturb the lipid phase of the cellular membrane may be correlated anti-MDR potency [29,48]. Additionally, the ability of IBC to kill MDR bacteria has been recently associated with the perturbation of the bacterial inner membrane by the chalcone [30]. In the present study, IBC was demonstrated to intercalate into model membrane composed of the zwitterionic lipid, phosphatidylcholine. The effects of the studied chalcone on the parameters of the main phospholipid phase transition included a decrease of transition temperature, enthalpy, and cooperativity. Based on an analysis of the thermotropic profiles of a vast number of chemicals, Jain and Wu [49] proposed the classification of the membrane perturbing effects observed by means of DSC into several types. According to this classification, IBC is likely to interact with the polar–apolar interface of the membrane, partially affecting the glycerol backbone region and the upper part of the acyl chain region. The present work constitutes the first study of the interaction of IBC with model membranes. Similar effects exerted on thermotropic properties of phosphatidylcholine membranes have been previously recorded for other prenylated polyphenols, 8-prenylnaringenin, xanthohumol, isoxanthohumol [32], licoisoflavone A, and 6,8-diprenylgenistein [33].

The existence of a relationship between anti-MDR potency and membrane-perturbing activity has been suggested for some MDR modulators [29,48], but it has not been identified thus far. In the case of Ver, the classical inhibitor of ABCB1, only limited effect on membrane properties was recorded. In several DSC studies performed with DMPC [50,51] and DPPC [52], Ver was observed to reduce slightly the transition temperature of the phospholipid, with no effect on transition enthalpy or cooperativity. The shape of calorimetric peaks was also hardly changed. The combination of solid state NMR and isothermal titration calorimetry allowed the conclusion that Ver strongly affected the head-group region of the phospholipid bilayer with only slight disordering of the fatty acid chains [53]. Additionally, Ver was found to interact preferentially with negatively charged phospholipids [53,54]. Overall, Ver was identified as a moderately active membrane-perturbing agent. The picture that emerged from the studies on interaction of Ver with model membranes pointed to the crucial role of its direct binding to ABCB1, but suggested that some lipid solubility was essential to be bound by the transporter [53].

Molecular modeling yielded additional information on IBC and its interaction with membranes. IBC was found to be relatively lipophilic, which facilitated its interaction with lipid bilayer. Additionally, an elongated shape of the molecule as well as its flexibility would facilitate intercalation of IBC within the lipid phase of cellular membrane. Another prenylated chalcone, xanthohumol, was previously discovered to affect model membranes to a greater extent as compared with prenylated flavonoids [32]. This notion was also supported by the results of PerMM simulation of membrane permeability. When looking at the visualization of the translocation pathway of IBC through membrane, it was noticed that the chalcone molecule rotated during permeation to keep its nonpolar parts closer to the membrane center and to place its polar atoms toward the membrane boundaries. The analysis of the free energy profile along the permeation pathway revealed the existence of two deep minima at the water−lipid interfaces and a local maximum close to the membrane center. Such profiles were previously reported to be characteristic of amphiphilic substances [55]. The calculation of membrane permeability coefficients suggested that IBC penetrated the simple model membranes with greater ease than natural membranes. Additionally, as demonstrated in the experimental part of the present work, IBC interacted with ABCB1 transporter and putatively also with other transporter proteins present in cellular membranes.

## 4. Materials and Methods

### 4.1. Materials

Two types of phosphatidylcholine, 1,2-dimyristoyl-sn-glycero-3-phosphocholine (DMPC) and 1,2-dipalmitoyl-sn-glycero-3-phosphocholine (DPPC), were purchased from Avanti Polar Lipids Inc. (Alabaster, AL, USA). IBC was from Alexis Biochemicals (Lausen, Switzerland). Ver, R123, Dox, and sulphorodamine B (SRB) were the products of Sigma-Aldrich (Poznan, Poland). All other chemicals were of analytical grade.

### 4.2. Cell Culture

Human colorectal adenocarcinoma HT29 cell line and its doxorubicin-resistant counterpart HT29/Dx were a kind gift of Prof. Chiara Riganti from Dept. of Oncology, University of Torino, Italy. HT29/Dx subclone was obtained by prolonged exposure to Dox as described previously [35] and maintained in medium containing 34 nmol/L doxorubicin. Madin-Darby Canine Kidney cells (MDCK) and MDCK cells expressing human ABCB1 (MDCK-MDR1) [56] were purchased from the Netherlands Cancer Institute (NKI-AVL, Amsterdam, The Netherlands). MDCK cells were cultured in DMEM and HT-29 cells in RPMI medium at 37 °C and 5% CO_2_. Both media were supplemented with 10% fetal bovine serum, l-glutamine, and antibiotics.

### 4.3. Cell Viability Assay

The cells were seeded on 96-well plates (15,000 cells/mL) 1 h before the experiment and incubated at 37 °C to attach firmly. Then, the cells were treated with the studied compounds for 48 h (at 37 °C, 5% CO_2_). Control wells contained medium only. Cytotoxicity of the solvent (DMSO) was also monitored and was found to be negligible. The further procedure was carried out as previously described [41]. All experiments were repeated 3 times.

### 4.4. Accumulation of Doxorubicin

Intracellular Dox accumulation was detected with fluorescence microscopy. HT29 and HT29/Dx cells were cultivated on 8-well µ-Slide microscopy chambers (Ibidi, Munich, Germany) for 48 h. For the experiment, a fresh portion of medium containing 50 µM Dox (plus 15 µM of IBC in treated samples) was added, and the cells were further incubated for 60 min at 37 °C. After incubation, the chambers were washed with PBS and with serum- and phenol red-free medium. The images were collected with a Nikon Eclipse TE2000-E microscope. Fluorescence was excited in the range 528–553 nm and collected in the range 578–633 nm.

### 4.5. Accumulation of Rhodamine 123

The cells were harvested and incubated (400,000 cells/mL) with the appropriate concentration of the studied compound (15 min, 25 °C). Then, R123 (5 μM) was added, and the cells were incubated for 60 min at 37 °C. After incubation, cells were washed and resuspended in PBS for flow cytometric analysis. Beckton Dickinson (Sunny Valley, ID, USA) FACSCalibur instrument equipped with a 488 nm argon laser was applied. Fluorescence was recorded via 530/30 nm band pass filter. A total of 5000 events were registered and analyzed with the use of Cell Quest^®^ software (Beckton Dickinson). Control samples were treated with medium only (no modulator). The influence of DMSO on the cells was also monitored and found to be negligible. Experiments were performed in triplicate. The fluorescence intensity ratio (FIR) was calculated from the following equation on the basis of measured fluorescence values (FL).
(1)$$\mathrm{FIR}=\frac{\left({\mathrm{FL}}_{\;\mathrm{MDCK}-\mathrm{MDR}1\;\mathrm{treated}})/({\mathrm{FL}}_{\mathrm{MDCK}-\mathrm{MDR}1\;\mathrm{control}}\right)}{\left({\mathrm{FL}}_{\mathrm{MDCK}\;\mathrm{treated}}\right)/\left({\mathrm{FL}}_{\mathrm{MDCK}\;\mathrm{control}}\right)}$$

### 4.6. Differential Scanning Calorimetry

IBC was dissolved in methanol to receive 5 mM stock solution. For each calorimetric measurement, 3 mg of lipid was dissolved in the appropriate amount of IBC solution to obtain specific drug:lipid mole ratios, i.e., 0.04, 0.06, 0.08. The mixtures were evaporated with a stream of nitrogen and dried under vacuum for at least 2 h to remove the solvent. Next, the thin lipid layer was hydrated with 15 µL of 20 mM Tris-HCl buffer (150 mM NaCl, 0.5 mM EDTA, pH = 7.4). Mixtures were heated above gel-liquid crystalline phase transition temperature of the lipid and vortexed to obtain homogenous dispersion. The samples were sealed in aluminum pans. Calorimetric measurements were performed using a DSC 600 microcalorimeter (Unipan, Warsaw, Poland). The samples were scanned at a rate 1 °C/min. Samples were scanned immediately after preparation. For each IBC:lipid mole ratio, at least 2 separate samples were prepared, and each sample was scanned at least 3 times. Data were analyzed with the software developed in our laboratory.

### 4.7. Molecular Modeling

Molecular modeling of IBC physico-chemical parameters was performed with Titan 1.0.8 software (Wavefunction, Irvine, CA, USA and Schrodinger, Portland, OR, USA) using semi-empirical AM1 approximation.

Calculation logP values and trans-bilayer energy profiles were performed with the publicly available PerMM web server (https://permm.phar.umich.edu/server, accessed on 21 May 2021) using the “global rotational optimization” option [38].

### 4.8. Data Analysis

Data represent the means ± standard deviation (SD) of 3 replications unless otherwise stated. Student’s *t*-test was used for checking statistical significance, and *p*-values less than 0.05 were considered to be significant.

## 5. Conclusions

IBC was shown to interact with ABCB1 transporter, interfering with its function. Most probably IBC was a substrate of ABCB1 and able to act as a competitive inhibitor in relation to other substrates of lower affinity to the transporter. Additionally, IBC was identified as an effective membrane-perturbing agent. According to our best knowledge, the interaction of IBC with lipid bilayer has been characterized here for the first time. Molecular modeling also pointed to the ability of IBC to interact with membranes. It was concluded that IBC interfered both with the lipid phase of cellular membrane and with MDR-associated transporter, and for this reason, its activity in MDR cancer cells was presumptively beneficial.

## Figures and Tables

**Figure 1 molecules-26-04637-f001:**
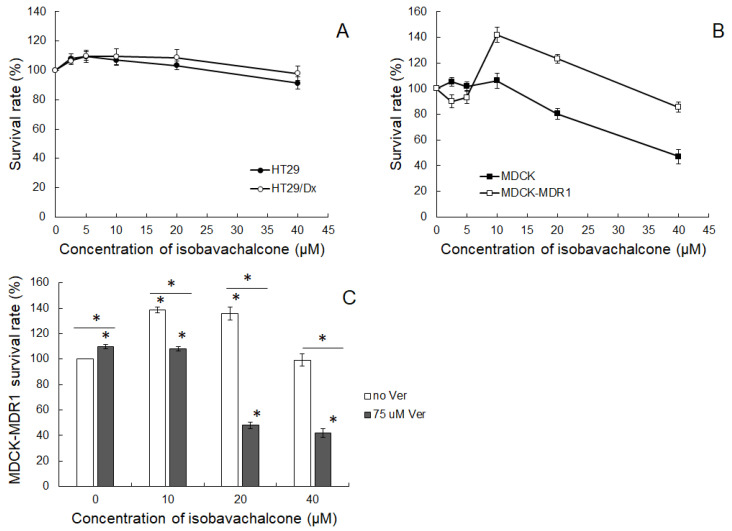
SRB cytotoxicity assay of IBC in HT29 and HT29/Dx (**A**), and MDCK and MDCK-MDR1 cells (**B**), and of IBC in combination with 75 μM Ver in MDCK-MDR1 cells (**C**). The means of three experiments ± SD are presented (* *p* < 0.05). Statistical significance was checked between the studied probes and controls (no IBC) and between probes containing only IBC and IBC combined with Ver. Statistical significance was achieved: in panel A for HT29 cells for 2.5, 5, and 40 μM IBC, in HT29/Dx for 2.5 and 5 μM IBC; in panel B for MDCK cells for 20 and 40 μM IBC, and in MDCK-MDR1 cells for 10, 20, and 40 μM IBC.

**Figure 2 molecules-26-04637-f002:**
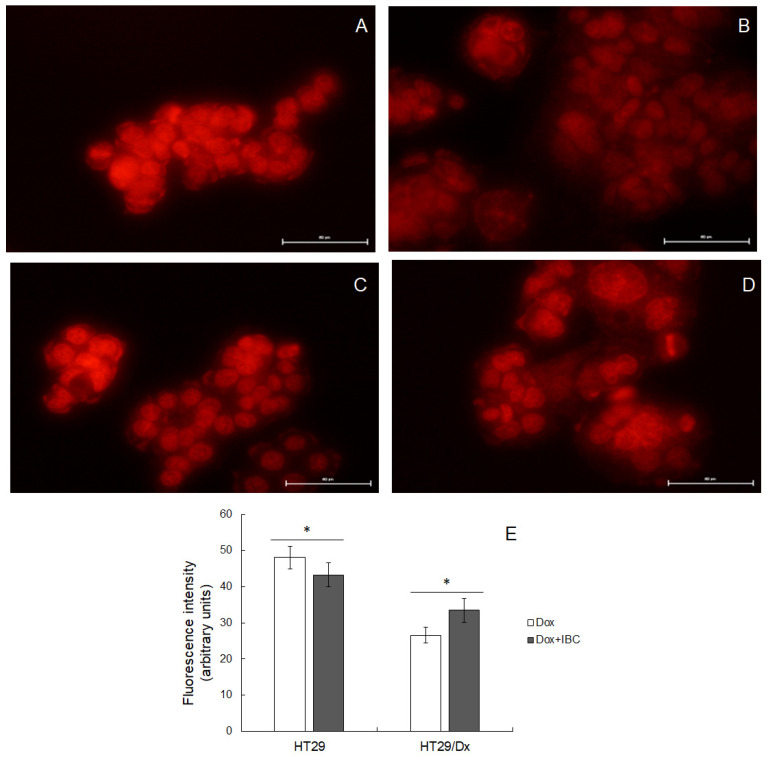
Microscopic images of Dox fluorescence observed in HT29 (**A**) and HT29/Dx cells (**B**) in the presence of IBC (at 15 μM) (**C**,**D**) for HT29 and HT29/Dx cells, respectively). Scale bar is 50 μm. Illumination conditions were the same for all images. Intensity of intracellular fluorescence as measured by ImageJ software (**E**) is presented as the mean fluorescence values ± SD measured in 20 representative cells. The statistically significant differences between IBC-treated and untreated cells were determined using Student’s *t*-test (* *p* < 0.05).

**Figure 3 molecules-26-04637-f003:**
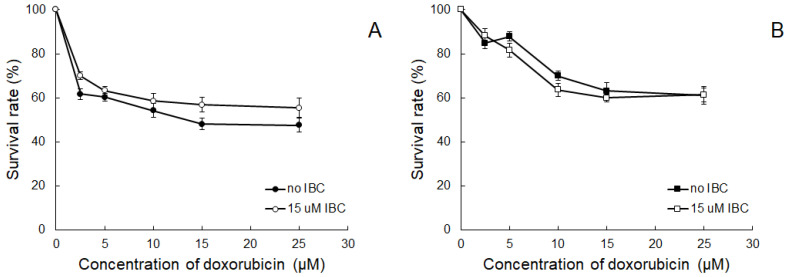
SRB cytotoxicity assay of Dox and Dox combined with IBC (at 15 μM) in HT29 (**A**) and HT29/Dx cells (**B**). The means of three experiments ± SD are presented. Statistically significant differences between cells treated only with Dox and Dox combined with IBC were not detected.

**Figure 4 molecules-26-04637-f004:**
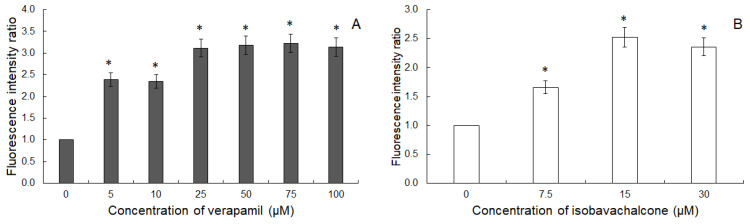
Intracellular accumulation of R123 in MDCK-MDR1 cells treated with Ver (**A**) and IBC (**B**). Means ± SD of three experiments are presented. The statistically significant differences from the untreated cells were determined using Student’s *t*-test (* *p* < 0.05).

**Figure 5 molecules-26-04637-f005:**
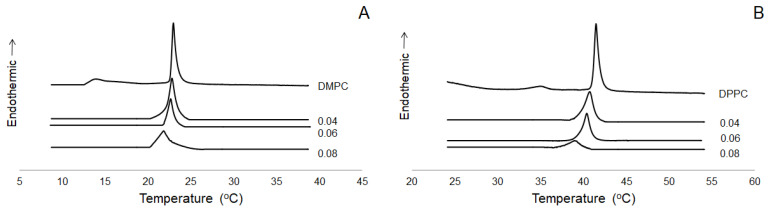
Thermograms of DMPC and IBC-DMPC mixtures (**A**), and DPPC and IBC-DPPC mixtures (**B**). Numbers in the figure represent IBC:lipid mole ratios. The thermograms were normalized to an equal amount of lipid for each profile.

**Figure 6 molecules-26-04637-f006:**
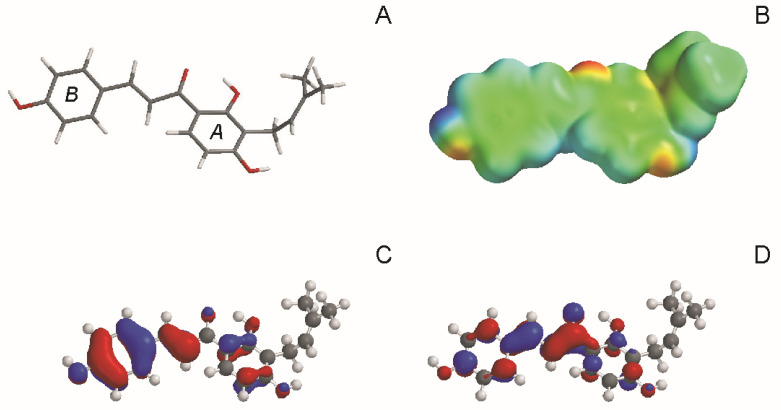
The ball and stick presentations of the optimized structure of IBC (**A**); electrostatic potential map of IBC (**B**) in aqueous solution (the blue color represents the positive region, and the red color represents the negative region); HOMO (**C**) and LUMO (**D**) orbitals of IBC in aqueous solution. All the images were obtained using Titan software.

**Figure 7 molecules-26-04637-f007:**
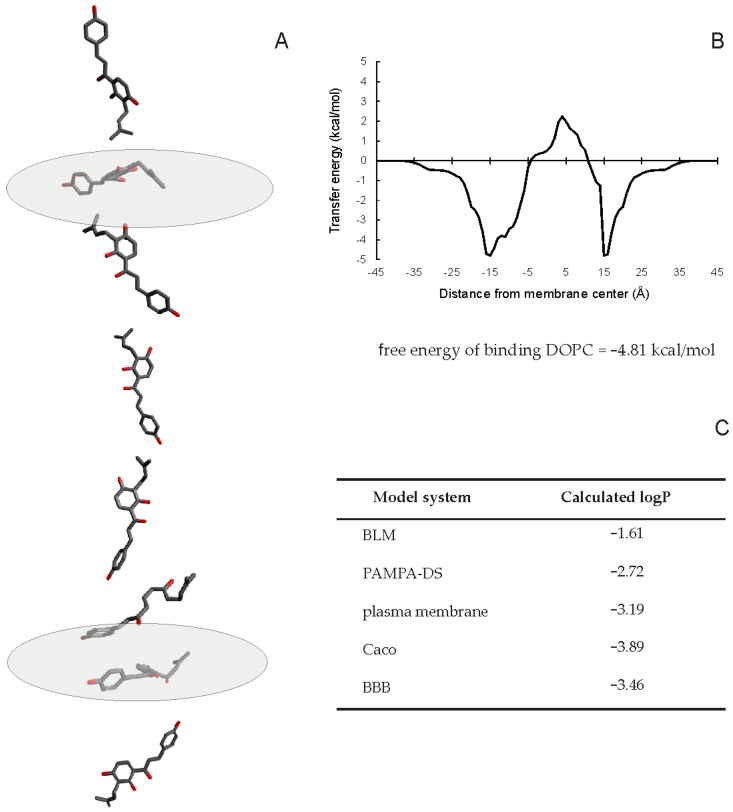
Spatial positions, optimized orientations (**A**), and transfer energy profile (**B**) calculated for IBC molecule as it crosses the DOPC bilayer; permeability coefficients calculated for various membrane models (**C**). Calculations s were performed by the publicly available PerMM web server using the “global rotational optimization” option.

**Table 1 molecules-26-04637-t001:** Main phospholipid phase transition parameters of phosphatidylcholines and IBC-phosphatidylcholine mixtures. Values are presented as means ± S.D.

Mole Ratio	Main Phase Transition Temperature [°C]	Transition Enthalpy [kJ/mol]	Half-Height Width [°C]
IBC:DMPC			
0	23.00 ± 0.09	26.00 ± 0.00	0.52 ± 0.05
0.04	22.66 ± 0.11	20.46 ± 3.21	0.70 ± 0.06
0.06	22.52 ± 0.18	15.20 ± 1.61	0.79 ± 0.10
0.08	21.86 ± 0.26	11.56 ± 1.64	0.98 ± 0.18
IBC:DPPC			
0	41.50 ± 0.11	36.50 ± 0.00	0.54 ± 0.04
0.04	40.73 ± 0.06	36.38 ± 4.52	1.05 ± 0.07
0.06	40.40 ± 0.27	37.58 ± 5.11	1.33 ± 0.51
0.08	39.64 ± 0.58	19.23 ± 5.04	1.65 ± 0.20

**Table 2 molecules-26-04637-t002:** Theoretical parameters obtained for studied compounds using Titan software.

Parameter	Isobavachalcone
MW	324.38
E_LUMO_ (eV)	−0.80
E_HOMO_ (eV)	−8.98
Energy gap (eV)	8.18
Octanol:water partition coefficient	4.19
Dipole moment (D)	1.57
Polarizability	64.92

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
