# Peer review of "Isobavachalcone as an Active Membrane Perturbing Agent and Inhibitor of ABCB1 Multidrug Transporter"

_molecules, 2021, doi:10.3390/molecules26154637_

Round 1

Reviewer 1 Report

The manuscript presents studies of great interest to the pharmaceutical and medical field regarding ABCB1 multidrug transporter inhibitors. 

  1. What were the criteria for choosing the membrane models (DMPC and DPPC) on which the studies were performed?
  2. For subsection 2.5, a comparison of the data obtained for IBC with those for a known inhibitor would provide additional data. 

Author Response

separate file uploaded

Reviewer 2 Report

The manuscript from Palko-Labuz et al investigated the cytotoxic, multidrug resistance, and ABCB1 transporter modulation effects of the active substance Isobavachalcone from medical plant, using two cancer cell lines. It is revealed that IBC might act as an ABCB1 transporter modulator and it is nicely shown by the authors through differential scanning calorimetry that IBC is a potent lipid bilayer modifier. The data are presented with sufficient analysis and the conclusions are reasonably supported by the observations. Overall, the text is logical and well written.

It is nice to see the difference that IBC killed more MDKC cells than MDKC-MDR1 cells which have overexpressed ABCB1 transporters. And the authors show that Ver the ABCB1 inhibitor make MDKC-MDR1 cells more vulnerable to IBC. It is surprising though that low concentrations of IBC not only did not kill MDKC-MDR1 cells but also significantly stimulated the growth of the MDKC-MDR1, could the authors comment on the stimulation effect in the discussion?

Some references in the Results section are missing, for example:

    Line 100, add reference for verapamil

    Line 114, add reference for Dox

    Line147, add reference for the Rho123 assay

Typos I identified in the text:

     Line 92, “ICB” should be “IBC”

     Line 117-118, “(Figure 2B and 2B, respectively)”, please fix

     Line 179, “form” seems should be “from”

Author Response

separate file uploaded
